A gene signature related to programmed cell death to predict immunotherapy response and prognosis in colon adenocarcinoma

Zheng Lei 1
Lu Jia 2
Kong Dalu 1 dalukong123@163.com
Zhan Yang 1 yangzhandr@163.com
1 Department of Colorectal Oncology, Tianjin Medical University Cancer Institute & Hospital, National Clinical Research Center for Cancer, Tianjin’s Clinical Research Center for Cancer, Key Laboratory of Cancer Prevention and Therapy, Tianjin Key Laboratory of Digestive Cancer , Tianjin , China
2 Department of Infection Management, Tianjin Medical University Cancer Institute & Hospital, National Clinical Research Center for Cancer, Tianjin’s Clinical Research Center for Cancer, Key Laboratory of Cancer Prevention and Therapy, Tianjin Key Laboratory of Digestive Cancer , Tianjin , China
Guan Fanglin
Electronic publication date: 2025 Feb 10
Publication date: 2025
Volume: 13
Electronic Location ID: e18895
Received 2024 Nov 1; Accepted 2025 Jan 3
Copyright: © 2025 Zheng et al.
Copyright year: 2025
Copyright holder: Zheng et al.
License: This is an open access article distributed under the terms of the Creative Commons Attribution License, which permits unrestricted use, distribution, reproduction and adaptation in any medium and for any purpose provided that it is properly attributed. For attribution, the original author(s), title, publication source (PeerJ) and either DOI or URL of the article must be cited.
License URL: https://creativecommons.org/licenses/by/4.0/

Keywords: Programmed cell death, RiskScore model, Colon adenocarcinoma, Immune microenvironment, Immunotherapy, Prognosis

Funding: Science and Technology Development Fund of Tianjin Education Commission for Higher Education 2020KJ133 Tianjin Key Medical Discipline (Specialty) Construction Project TJYXZDXK-009A This study was supported by the Science and Technology Development Fund of Tianjin Education Commission for Higher Education (2020KJ133) and the Tianjin Key Medical Discipline (Specialty) Construction Project (TJYXZDXK-009A). The funders had no role in study design, data collection and analysis, decision to publish, or preparation of the manuscript.

==============================
Background

Tumor development involves the critical role of programmed cell death (PCD), but the correlation between colon adenocarcinoma (COAD) and PCD-related genes is not clear.

Methods

Subtyping analysis of COAD was performed by consensus clustering based on The Cancer Genome Atlas (TCGA), with the AC-ICAM queue from the cBioportal database as a validation set. Immune infiltration of the samples was evaluated using CIBERSORT and Microenvironment Cell Populations (MCP)-counter algorithms. Patients’ immunotherapy response was predicted by the TIDE and aneuploidy scores. Pathway enrichment analysis was conducted with gene set enrichment analysis (GSEA). A RiskScore model was established with independent prognostic PCD-related genes filtered by Cox regression analysis. The mafCompare function was used to compare the differences in mutation rates of somatic genes. Wound healing, transwell assays and Flow cytometer were applied to measure the cell migration, invasion and apoptosis.

Results

The patients were grouped into S1 and S2 subtypes based on a total of 21 PCD genes associated with the prognostic outcomes of COAD. Specifically, patients of S1 subtype were mainly related to the pathway activation in tumor invasion and deterioration and had a worse prognosis. A RiskScore model was established based on six prognostic genes, including two protective genes (ATOH1, ZG16) and four risk genes (HSPA1A, SEMA4C, CDKN2A, ARHGAP4). Notably, silencing of CDKN2A inhibited the activity of migration and invasion and promoted apoptosis of tumor cells. Based on the RiskScore model, the patients were grouped into high- and low-risk groups. Independent prognostic factors, namely, Age, pathologic_M, pathologic_stage, and RiskScore, were integrated to develop a nomogram with strong good prediction performance. High-risk group had high-expressed immune checkpoint genes and higher TIDE scores, showing a strong immune escape ability and less active immunotherapy response. Compared to the low-risk group, TP53 exhibited a higher rate of somatic mutation in the high-risk group.

Conclusion

We constructed a RiskScore model with six PCD-related genes for the prognostic assessment of COAD, providing a valuable insight into the exploration of new targets for the prognostic improvement in COAD.

Introduction

Colon cancer is closely related to dietary habits and is also one of the crucial factors leading to cancer-correlated mortality (Ma et al., 2021; Hu et al., 2021). The American Cancer Society showed that the incidence and death rate of colon cancer are 10.2% and 9.2%, respectively, posing a serious threat to human health (Wen et al., 2019). Colon adenocarcinoma (COAD) as the primary pathological subtype of colon cancer is mainly treated with surgical local resection, radiotherapy, chemotherapy, targeted therapy, and immunotherapy (Yin & Guo, 2021; Sun et al., 2023), which have significantly improved the overall survival (OS) of COAD patients. However, the relative survival rate of COAD is still low due to tumor recurrence and metastasis that limit the therapeutic efficacy. Hence, mining effective therapeutic targets for COAD has become an urgent task to improve the COAD prognosis.

Programmed cell death (PCD) is a type of apoptosis that plays a critical role in cell development, tissue homeostasis, and elimination of damaged cells (Shi et al., 2023; Karati & Kumar, 2024). PCD is regulated by many genes and interacts with numerous signaling pathways (Pan et al., 2022). Study confirmed that PCD has 12 cell death modes (Tang et al., 2019) and each plays a critical part in the malignant progression and metastasis of tumors (Waghela et al., 2021). In general, cells cannot replicate and metastasize permanently. Suppression of PCD will disrupt normal cell death, leading to uncontrolled cell proliferation and transformation into cancer cells (Dai, Wang & Zhang, 2021) and ultimately into the formation of various diseases such as COAD (Chen et al., 2022). PCD has now been considered as an effective approach to the design of anti-tumor drugs (Zhou et al., 2024; Akhtar et al., 2023). At present, many compounds have been proven to be able to suppress tumor cell proliferation. For example, aldose reductase inhibition suppresses COAD cell viability via modulating the expression of mediated programmed cell death four expression (PDCD4) mediated by miR-21 (Saxena et al., 2013). Methanol extract of Coleus amboinicus (Lour) inhibits the proliferation of COAD cell WiDr and stimulates PCD (Laila et al., 2020). Panaxadiol suppresses the expression of PCD-ligand 1 and tumor proliferation by hypoxia-inducible factor-1α and STAT3 in COAD cells (Wang et al., 2020; Li et al., 2022). However, the association between PCD and COAD is less investigated.

This work explored the role of genes related to PCD in immunotherapy response and prognostic prediction for patients suffering from COAD. Firstly, based on PCD-related genes, consensus clustering analysis categorized two subtypes of COAD, from which differentially expressed genes (DEGs) were screened. Differences in pathways enriched in the two subtypes were compared by Gene Set Enrichment Analysis (GSEA). Finally, six significant genes were determined by LASSO Cox and stepwise regression and used to establish a robust RiskScore model for the prognostic assessment for COAD. The combination of RiskScore and clinical pathological features also showed a strong performance in external datasets. This study improved the understanding of the relationship between PCD and the development of COAD, providing potential treatment targets for the cancer.

Materials and Methods

Data collection

Single nucleotide variation (SNV) mutation information, the expression data in transcripts per million (TPM) format, and clinical follow-up information of COAD patients were collected from the TCGA database (https://portal.gdc.cancer.gov/) (Shahrajabian & Sun, 2023). The RNA-Seq expression profile was log2-transformed and Ensembl was converted to gene symbol. The SNV mutation information was processed by mutect2. A sum of 427 COAD samples with survival time longer than 30 days were obtained (Table S1). In addition, the cBioportal database (https://www.cbioportal.org/) was accessed to acquire the RNA-seq expression and survival data of COAD in AC-ICAM cohort (validation set), and a total of 341 tumor samples with a survival time longer than 30 days were retained. Based on a previous study, 1,254 PCD-associated genes were extracted for our follow-up study (Zou et al., 2022).

Consensus clustering analysis

Firstly, prognostically relevant genes were selected from a total of 1,254 PCD-related genes by univariate Cox analysis. Then, consensus clustering was conducted using ConsensusClusterPlus R package (parameters: clusterAlg = “pam” and distance = “pearson”) (Chen et al., 2022). Sampling was repeated 500 times at a sampling ratio of 80% each time. The threshold was between two and 10 to decide the optimal number for the clustering.

Evaluation of immune microenvironment and immunotherapy response

CIBERSORT (https://cibersort.stanford.edu/) was used to quantify the density of 22 types of immune cells in COAD samples (Chen et al., 2018). MCP-counter was used to quantify the abundance of three types of stromal cells and eight types of immune cells based on gene expression matrix (Becht et al., 2016).

Immunotherapy benefits to different risk groups were predicted applying TIDE (http://tide.dfci.harvard.edu/) and aneuploidy scores. Potential immune checkpoint blockade therapy responses in COAD patients were evaluated by TIDE, with a lower TIDE score suggesting lower possibility of immune escape and a more active immunotherapy response (Wang et al., 2022). Besides, the aneuploid score of TCGA samples was obtained from a published article (Huang & Fu, 2019), with higher aneuploidy score indicating higher possibility of evading immune surveillance and less active immunotherapy response.

GSEA

Pathway enrichment analysis was conducted using GSEA and the pathways with Padj < 0.05 were defined as differential pathways. The HALLMARK pathway gene set was collected from the molecular feature database (https://www.gsea-msigdb.org/gsea/msigdb/) as a background gene set.

Identification of DEGs

The DEGs in S1 subtype and S2 subtype were selected with the limma package under |log2FoldChange(FC)| > log2(1.5) and Padj < 0.05 (Song et al., 2023). The clusterProfiler R package (Lin et al., 2024) was employed in Kyoto Encyclopedia of Genes and Genomes (KEGG) enrichment analysis, and the pathways with Padj < 0.05 were considered as significantly enriched (Song et al., 2023).

Development of a RiskScore model and validation

The number of prognostically relevant genes screened by performing univariate Cox proportional hazards regression (p < 0.05) were reduced by conducting LASSO Cox regression analysis in the R package “glmnet” (Li, Lu & Yin, 2022). Key genes and correlation coefficients were obtained through stepwise regression, and the formula of the RiskScore was as follow (Chu et al., 2023):

RiskScore=∑βi∗ExPi.

βi and ExPi represent the coefficient of gene in Cox regression model and the gene expression value, respectively.

Based on the median RiskScore, low- and high-risk groups of COAD patients were classified. Then, the receiver operating characteristic (ROC) curve was developed with “timeROC” package (Lu et al., 2022). The RiskScore model was also validated in AC-ICAM cohort.

Construction of a nomogram and validation

Based on the results of univariate and multivariate Cox regression analyses, the prognostic nomogram was established by integrating Age, pathologic_M, pathologic_stage, and RiskScore using rms package (Liu et al., 2023). The accuracy and reliability of the nomogram were validated by the calibration curve and the decision curve analysis (DCA), respectively.

Genomic feature analysis

The data of SNV somatic mutation was processed using “maftools” R package (Mayakonda et al., 2018), and the differences in somatic gene mutation rates of the two different cohorts were compared using the mafCompare function.

Cell culture

Dulbecco’s modified Eagle medium (DEME) (Gibco-Thermo Fisher Scientific, Waltham, MA, USA) that supplemented with 10% fetal bovine serum (FBS), and 1% Penicillin and Streptomycin (BI, Hamagshimi, Israel) was used to culture human normal colonic epithelial cells (NCM460) obtained from Baidi Biotech Ltd. (Hangzhou, China) and human colon adenocarcinoma cells (SW1116) purchased from the Kalang biotechnology company (Shanghai, China) at 37 °C and 5% CO2.

RT-qPCR

Cells of 80–90% growth were treated by trypsin and washed with PBS for collection. Then, the TRIzol reagent Kit (Vazyme, Nanjing, China) was added to isolate total RNA, which was reverse-transcribed and quantified using a commercial kit (Vazyme, Nanjing, China) for synthesizing cDNA based on the manufacturer’s specification. Finally, the qPCR reaction system was mixed by the SYBR Green PCR Master Mix (Invitrogen) for qRT-PCR on LightCycler 96 (Roche), and the ΔCT, ΔΔCT value, 2−ΔΔCT value were calculated for each sample (Kang et al., 2022). The specific primers were shown in Table S2.

Wound healing and transwell assay

The si-CDKN2A (Forward: 5′-GCGCGGAGCCCAACTGCG-3′; Reverse: 5′-AGCGTCGTGCACGGGTCG-3′) obtained from the Sangon Biotech company (Shanghai, China) was transfected into cells by adding Lipofectamine 3000 (Invitrogen). The SW1116 cells with si-CDKN2A silencing was used for the wound healing assays, and 4 × 106 cells were seeded into the 12-well plate containing DMEM. After reaching 95% confluence, the tip of a 100 μl pipette was employed to scratch cell layers, ensuring the same width of each wound. Next, the cells were rinsed in PBS reagent and transferred into complete medium with 1% FBS for further incubation at of 37 °C with 5% CO2. The width of the cell injuries was measured with an inverted microscope immediately after the scratch and 48 hours (h) later.

For transwell test, the 12-well plates with 8-μm pore size chamber (Corning) were used. The cell suspension (200 μl) was supplemented into the upper chamber of the Matrigel-coated serum-free transwell, while the lower chamber was added with 500 μl of complete medium with 10% FBS. After incubating the transwell setup in a 5% CO2 environment for 48 h at 37 °C, 0.1% crystal violet solution were added to dye the cells in the lower chamber for 15 minutes (min) and the number of cells were counted with an inverted microscope (Leica, Wetzlar, Germany).

Flow cytometry

The harvested cells were resuspended in PBS at a density of 1 × 105 cells per 200 μL and subsequently stained with Annexin V-FITC and PI solution (C1062S, Beyotime, China) on ice for 30 min in the dark. Following washing the cells with PBS, BD FACSAria III flow cytometer (BD, Franklin Lakes, NJ, USA) was utilized to analyze the samples (Tian et al., 2023).

Statistical analysis

All statistical data were analyzed in R language (version 4.2.0 and Graph Prism 8.0). Differences between two-group continuous variables were calculated with Wilcoxon rank-sum test. The statistics of experimental data were performed by student’s t test. The correlation was calculated using Spearman method. Survival difference in different groups was compared by plotting Kaplan-Meier (KM) curves with the log-rank test and p < 0.05 represented statistically significant.

Results

Two molecular subtypes of COAD were identified using PCD-related genes

Firstly, 21 PCD-related prognostic genes for COAD were obtained from TCGA and AC-ICAM cohorts (Figs. 1A and 1B). Based on consensus clustering analysis, 427 COAD samples in TCGA were divided into two subtypes (S1 and S2) when consensus matrix k = 2 (Fig. 1C). Specifically, S2 subtype showed a relatively favorable prognosis, while S1 subtype had a worse prognosis (Figs. 1D and 1E). Further analysis revealed that the protective genes (IL13, GLB1, IL13RA2, SLC39A8, FAS) were high-expressed in S2, while the risk genes were high-expressed in S1 (Figs. 1F and 1G). In addition, compared with S2 subtype, the distribution of clinical and pathological features in S1 and S2 subtypes in the TCGA dataset revealed that more patients in S1 subtype were in the pathologic_T3, pathologic_T4, pathologic_N1, pathologic_N2, pathologic_M1, pathologic_stage III, and pathologic_stage IV (Fig. 1H).

Figure 1 Consensus clustering analysis based on PCD-related genes.

(A, B) PCD genes associated with prognosis in TCGA and AC-ICAM cohorts. (C) Heatmap of TCGA consistency clustering. (D, E) Survival difference between two different subtypes of patients in the TCGA and AC-ICAM cohorts. (F, G) Expression levels of PCD genes related to prognosis in TCGA-COAD and AC-ICAM cohorts. (H) Distribution of different subtypes of colon cancer in clinical variables.

Differences in immune microenvironment and pathway characteristics between S1 and S2 subtypes

Next, the immune microenvironment between S1 subtype and S2 subtype was compared. CIBERSORT analysis showed that the abundance of T cell CD4+ memory, macrophage M1, and mast cell resting was high in S2, while S1 had more Tregs and macrophage M0 (Fig. 2A). MCP-counter analysis indicated that the immune cell scores of natural killer (NK) cell and cytotoxicity score in S2 were remarkably higher than those in S1 and the cancer-associated fibroblast and endothelial cell were high in S1 group (Fig. 2B). This indicated that the S2 subtype may have a stronger anti-tumor immune microenvironment, while the S1 subtype may have an immunosuppressive environment. GSEA showed that S1 subtype were predominantly enriched in pathways related to tumor invasion and progression, including myogenesis, angiogenesis, and epithelial mesenchymal transition (EMT) (Fig. 2C). In contrast, S2 subtype was notably enriched in the pathways linked with cell proliferation, including G2m Checkpoint, MYC Targets V1, and E2f Targets (Fig. 2D).

Figure 2 Analysis of immune and pathway characteristics between subtypes.

(A) Difference in CIBERSORT immune infiltration between different subtypes in TCGA. (B) Difference in MCP-counter immune scores between different subtypes in TCGA. (C) GSEA analysis of S1 subtype in TCGA. (D) GSEA analysis of S2 subtype in TCGA. **** means p < 0.0001; ** means p < 0.01; * means p < 0.05; ns means not significant.

Identification of DEGs between S1 and S2 subtypes

DEGs analysis screened a total of 90 downregulated and 394 upregulated genes from S1 and S2 subtypes in the TCGA cohort (Fig. 3A). The heatmap displayed the top 50 DEGs (Fig. 3B). KEGG analysis showed that the upregulated genes were principally enriched in TGF−β signaling pathway, ECM−receptor interaction, and proteoglycans in cancer (Fig. 3C), while the downregulated expression genes were largely enriched in the chemokine signaling pathway, IL−17 signaling pathway, and cytokine−cytokine receptor interaction (Fig. 3D).

Figure 3 Analysis of differentially expressed genes (DEGs) between subtypes.

(A) Volcano map of DEGs between subtypes in TCGA cohort. (B) Heat map of DEGs between subtypes in TCGA cohort. (C,D) Results of KEGG enrichment analysis of DEGs between subtypes in TCGA cohort.

RiskScore model with a strong prognostic prediction performance

A total of 484 DEGs were subjected to univariate Cox proportional hazards regression. Then, the gene number was further narrowed using LASSO Cox and stepwise regression (Fig. 4A). Six key genes were obtained to establish a RiskScore model of “RiskScore=(0.208*HSPA1A) + (0.273*SEMA4C)+(−0.087*ATOH1)+(0.154*CDKN2A)+(0.163*ARHGAP4) + (−0.073*ZG16)”, with ATOH1 and ZG16 as the protective genes (hazard ratio < 1) and SEMA4C, HSPA1A, ARHGAP4, and CDKN2A as the risk genes (hazard ratio>1) (Figs. 4B and 4C). COAD patients were divided into high- and low-risk groups by the median RiskScore value. Furthermore, the ROC for the TCGA cohort showed that the RiskScore model had 1-, 2-, 3-, 4-, and 5-year AUC of 0.69, 0.7, 0.69, 0.72, and 0.69, respectively (Fig. 4D). In TCGA cohort, the low-risk group had better OS, disease-specific survival (DSS), and progression-free interval (PFI) in comparison to the high-risk group (Figs. 4E–4G). Moreover, the ROC analysis showed that the RiskScore model was highly robust in predicting 1- to 5-year survival in the AC-ICAM cohort, with AUC values of 0.72, 0.61, 0.6, 0.58, and 0.6 (Fig. 4H). Survival analysis in AC-ICAM cohort also showed a low OS rate and poor prognosis in high-risk group (Fig. 4I). Additionally, principal component analysis (PCA) revealed an obvious boundary between the high- and low-risk groups in the two cohorts (Fig. 4J). Differential expression analysis of the model genes demonstrated that ATOH1 and ZG16 were high-expressed protective factors in the low-risk group, while SEMA4C, HSPA1A, ARHGAP4, and CDKN2A were high-expressed risk factors in the high-risk group (Fig. 4J).

Figure 4 Establishment of the risk model.

(A) Results of narrowing the gene range through LASSO. (B) Model genes and their coefficients. (C) Multi factor forest map of model genes. (D) ROC curves of risk model for 1–5 years in TCGA cohort. (E–G) Differences in overall survival (OS), disease-specific survival (DSS), and progression-free interval (PFI) between high and low-risk groups in TCGA cohort. (H, I) ROC curves for 1–5 years and survival difference of AC-ICAM validation cohort model. (J) Gene expression patterns and PCA analysis scatter plots of high and low-risk patients in TCGA and AC-ICAM cohorts.

A nomogram was developed with a strong predictive performance

Analysis on the clinical grades in TCGA cohort showed that the RiskScore was higher in the pathologic_stage III and IV than the pathologic_stage I and II, higher in the pathologic_T3 and T4 than the pathologic_T1 and T2, higher in the pathologic_N1 and N2 than the pathologic_N0, higher in the pathologic_M1 than the pathologic_M0 (Fig. 5A). The distribution of clinical and pathological features between the two risk groups of COAD patients demonstrated that more patients were in the pathologic_T4, pathologic_N2, pathologic_M1, and pathologic_ stage IV in high-risk group (Fig. 5B). Moreover, univariate and multivariate Cox regression analysis on the clinical features and RiskScore confirmed that Age, the RiskScore, pathologic_M, pathologic_stage were independent factors for evaluating the prognosis of COAD (Figs. 5C and 5D). Then, a nomogram was developed. It was observed that RiskScore exhibited the greatest impact on the prediction of survival rate (Fig. 5E). Furthermore, the predicted calibration curves for 1-, 3-, and 5-year survival by the nomogram were close to the standard curves (Fig. 5F), indicating that the prediction by the nomogram was accurate. Additionally, the reliability of nomogram was assessed by DCA, which showed that the calibration curves indicative of the benefits of RiskScore and nomogram were notably higher than the extreme curves (Fig. 5G).

Figure 5 Construction and assessment of a survival nomogram model.

(A) RiskScore difference among clinical grades in TCGA. **** means p < 0.0001; * means p < 0.05. (B) Distribution of clinical variables between different risk groups for colon cancer. (C, D) Single factor and multi factor results of RiskScore and clinical features. (E) Nomogram established by combining RiskScore and clinical features. (F, G) Calibration curve and decision curve of nomogram.

High-risk group exhibited strong immune escape ability and poor immunotherapy response

Compared to the high-risk group, CIBERSORT analysis showed that the abundance of mast cell resting, T cell CD4+ memory, and B cell Plasma was higher in low-risk group (Fig. 6A). MCP-counter analysis revealed the strongest positive correlation between the RiskScore and cancer-associated fibroblasts (CAFs) (Fig. 6B). Moreover, comparison on the expressions of seven immune checkpoint genes indicated that the expression of four immune checkpoint genes (PDCD1, HAVCR2, TIGIT, and LAG3) was higher in high-risk group (Fig. 6C), showing that high-risk COAD patients had strong immune escape ability and limited benefits from immune checkpoint inhibitors. Additionally, the RiskScore and TIDE score were positively correlated, with a correlation coefficient R of 0.39 (Fig. 6D). The number of respondents was higher in low-risk group (39.72%) than in high-risk group (26.29%) (Fig. 6E), showing a greater immunotherapy benefit to low-risk COAD patients. The aneuploid score was higher in high-risk group than that in low-risk group (Fig. 6F), which showed that tumor cells in high-risk patients may have greater genomic instability and may be more likely to evade immune surveillance and become resistant to treatment.

Figure 6 Evaluation of TME abnormality and immunotherapy response in risk grouping.

(A) The abundance difference in CIBERSOER immune infiltration between TCGA risk groups. (B) Spearman correlation between RiskScore and MCP-counter immune score in TCGA cohort. (C) Expression difference in immune checkpoint between risk groups in TCGA cohort. (D) Correlation between RiskScore and TIDE score in TCGA cohort. (E) Distribution of immune response between risk groups in TCGA cohort. (F) Difference in Aneuploid score between risk groups in TCGA cohort. **** means p < 0.0001; *** means p < 0.001; ** means p < 0.01; * means p < 0.05; ns means not significant.

Differences in GSEA and somatic mutation between the two risk groups

Pathways such as EMT, angiogenesis, and interferon-γ response were activated in high-risk group but fatty acid metabolism was inhibited (Fig. 7A). Further analysis showed that TP53 exhibited a particularly higher somatic mutation rate in high-risk group than in low-risk group (Fig. 7B).

Figure 7 Differences in GSEA and somatic mutation between risk groups.

(A) Difference in GSEA activation pathways between risk groups. (B) Difference in somatic mutations between risk groups.

Expression of the model genes validated in vitro assay

The results of RT-qPCR showed that HSPA1A, SEMA4C, ARHGAP4 and CDKN2A were significantly overexpressed in the SW1116 cells, whereas ATOH1 and ZG16 were significantly downregulated in the cells (Fig. 8A). As CDKN2A was a high-risk gene, its expression level was significantly elevated in high-risk group, which showed a poor prognosis. Hence, CDKN2A was chosen as the subject of functional validation due to such a strong correlation. The wound healing and transwell assay revealed that silencing CDKN2A significantly impaired the tumor cell migration and invasion (Figs. 8B–8E). The apoptosis rate of cells was significantly elevated after CDKN2A knockdown in comparison to a low apoptosis rate in the control group, suggesting that CDKN2A may play a crucial role in the survival of colorectal cancer cells by inhibiting apoptosis (Fig. 8F). These results further supported the potential oncogenic function of CDKN2A in COAD progression.

Figure 8 The vitro assay of model genes.

(A) The expression difference of model among normal and tumor cells. (B) The wound closure difference among si-NC and si-CDKN2A groups. (C) The wound healing assays. (D) The cell number difference among si-NC and si-CDKN2A groups. (E) Imaging of trans-well assays. (F) apoptosis was determined by Flow cytometer. **p < 0.01, ****p < 0.0001.

Discussion

COAD is a frequent malignant tumor in the world (Peng et al., 2024). Recently, gene signatures related to PCD have been increasingly developed to evaluate the prognosis and therapeutic responses in COAD (Wei et al., 2023). Based on PCD-related genes, this study divided patients with COAD into two subtypes with distinct prognosis, immune microenvironment, and pathway characteristics. Then, 6 PCD-related genes (two protective genes and four risk genes) were identified from the DEGs between the two subtypes and used to create a RiskScore model for the prognostic evaluation of COAD. The model could classify COAD patients into low- and high-risk groups, exhibiting a strong robustness in predicting the immunotherapy response and prognosis for COAD patients.

As PCD can notably influence the development of COAD, PCD-related genes may have the potential to serve as prognostic signature for COAD (Chen et al., 2022). Previous study (Peng et al., 2024) established a PCDscore system based on six PCD-correlated genes for COAD, and it was found that high-PCDscore group with low CD4+ T cell infiltration and high stromal score exhibited poor prognosis. In the study of Zhu, Kong & Xie (2021), COAD patients were classified into two subtypes using 15 ferroptosis-correlated genes, with cluster2 showing lower mutation burden and expressions of checkpoint genes and more favorable survival. Then, a prognosis model was constructed using the 15 ferroptosis-related genes. Furthermore, Zhuang et al. (2021) defined two molecular clusters (MC) of COAD and observed that patients with MC1 exhibited a poor OS. A 13 OS-related prognostic signature was developed, with high-risk patients exhibiting a poorer OS in comparison to those in low-risk group (Zhuang et al., 2021). In our study, COAD samples were divided into S1 and S2 using PCD-related genes, and the prognostic outcomes of S1 patients were more unfavorable. Subsequently, six PCD-related prognostic genes, including two protective genes (ATOH1, ZG16) and four risk genes (HSPA1A, SEMA4C, CDKN2A, ARHGAP4), were obtained to create a RiskScore model. COAD patients were separated into high- and low-risk groups, and it was found that the low-risk group had high expressions of protective genes, while the high-risk group had high expressions of risk genes.

Tumor microenvironment (TME) can regulate the development of COAD through complex interactions via the exchange of molecular information (Cheng et al., 2024). Zabeti Touchaei, Vahidi & Samadani (2024) reported that NK cell, T cell CD4+, T cell CD8+, and macrophages were dense in the colon microenvironment and regulated the mechanism of COAD development, immune evasion, and sensitivity to standard chemotherapy (Song et al., 2023). In our study, we found that the abundance of T cell CD4+ memory, mast cell resting, and macrophage M1 was high in S2 subtype, while that of Tregs and macrophage M0 was high in S1 subtype. The immune cell scores of NK cell and cytotoxicity score were remarkably higher in S2 subtype than in S1 subtype, while the CAFs and endothelial cell were high in S1 subtype. CD4+T cells, NK cells, macrophage M1, and CD8+T cells are anti-tumor immune cells, while Tregs and macrophage M2 are pro-tumor immune cells (Lv et al., 2022; Liu et al., 2021; Gu et al., 2024). Moreover, Wang reported that the activated mast cells contributed to the development of lung adenocarcinoma, while the resting mast cells exhibited an anti-tumor potential (Wang et al., 2020). Tregs in the TME could facilitate tumor development by suppressing anti-tumor immune responses and supporting transformed cell survival (Glasner & Plitas, 2021). Our results demonstrated that the S1 showed an immunosuppressive environment, while the S2 exhibited an active anti-tumor immune microenvironment. This was consistent with the poor prognosis of S1 subtype and favorable prognosis of S2 subtype.

Furthermore, in vitro assay also showed that the four risk genes (HSPA1A, SEMA4C, CDKN2A, ARHGAP4) were overexpressed in the tumor cells. ATOH1 has been considered as a tumor-suppressive gene silenced in gastrointestinal tumors in which it was often expressed at a high level (Yang et al., 2018; Fukushima et al., 2015). COAD patients with a high ATOH1 expression tended to have a strong immunogenicity and favorable prognosis (Mou et al., 2022). ZG16 is a mammalian lectin-like protein that produces at a high level in human colon. It has been proven that the absence of ZG16 could induce the occurrence and progression of colorectal cancer (CRC), indicating that ZG16 may have an anti-tumor effect (Meng et al., 2021). In our study, the expression of protective genes (ATOH1, ZG16) in patients with a low risk was higher than in those with a high risk. CRC patients with lower expression of HSPA1A, a heat shock protein, had higher OS rates (Ding et al., 2022; Jiang et al., 2021). SEMA4C, a semaphorin, can promote tumor development and serve as a candidate treatment target for cancers. SEMA4C is an independent prognostic predictor in CRC patients as high-expressed SEMA4C promoted the EMT and was predictive of poor prognosis in CRC (Hou et al., 2020). CDKN2A as a cell cycle-associated protein may facilitate CRC cell metastasis by inducing EMT. High-expressed CDKN2A is related to a poor prognosis in CRC (Dong et al., 2023) through affecting the migration and invasion of the tumor cells. ARHGAP4 is a Rho GTPase-activating protein (Sera et al., 2022). CRC patients with high-expressed ARHGAP4 usually exhibited a poor prognosis, suggesting that ARHGAP4 may be a prognostic biomarker for CRC (Fu et al., 2022). Similarly, our study showed that the risk genes (HSPA1A, SEMA4C, CDKN2A, ARHGAP4) were high-expressed in high-risk group in which EMT, angiogenesis, and interferon-γ response pathways were activated. In addition, it was also found that some immune checkpoint genes (PDCD1, HAVCR2, TIGIT, LAG3) had high expression in high-risk group, which was related to a higher TIDE and aneuploidy scores than in the low-risk group. The above results manifested that COAD patients in low-risk group had a favorable prognosis and were likely to benefit from immunotherapy, while high-risk COAD patients had stronger immune escape possibility and limited benefits from taking immunotherapy. These findings supported that PCD-related prognostic genes identified by this study could serve as biomarkers for COAD management.

However, the current study still had certain limitations. Firstly, the RiskScore model established based on TCGA database was not validated using clinical cohort samples. Secondly, the mechanism underlying the effects of PCD-related genes on the development of COAD still required experimental validation. In the future, clinical validation and relevant experiments such as molecular biology experiments, cell-based and animal experiments should be supplemented to confirm our current discoveries.

Conclusion

Taken together, the current study developed a RiskScore model using PCD-related genes for evaluating the immunotherapy response and prognosis for COAD patients. It was found that high-risk patients showed stronger immune escape and worse prognosis, and that genes such as CDKN2A played an important role in tumor cell migration, invasion and apoptosis. The current findings provided potential targets to facilitate the diagnosis and treatment of COAD and improved the understanding on the mechanisms of cell death in the development of COAD.

Supplemental Information

Supplemental Information 1 Information on clinical samples of COAD patients obtained based on TCGA.

Supplemental Information 2 Sequence of target gene primer pairs for RT-qPCR.

Supplemental Information 3 MIQE checklist.

Abbreviation

AUC area under ROC curve

CAFs cancer-associated fibroblasts

CIBERSORT Cell-type Identification by Estimating Relative Subsets of RNA Transcripts

COAD colon adenocarcinoma

CRC colorectal cancer

DCA decision curve analysis

DEGs differentially expressed genes

DSS disease-specific survival

EMT epithelial mesenchymal transition

FC fold change

GSEA gene set enrichment analysis

KEGG Kyoto Encyclopedia of Genes and Genomes

LASSO Least absolute shrinkage and selection operator

MC molecular clusters

MCP-counter Microenvironment Cell Populations-counter

NK natural killer

OS overall survival

PCA principal component analysis

PCD programmed cell death

PFI progression-free interval

ROC receiver operating characteristic

SNV single nucleotide variation

TCGA The Cancer Genome Atlas

TIDE Tumor Immune Dysfunction and Exclusion

TME tumor microenvironment

TPM transcripts per million

Tregs T cell regulatory

Additional Information and Declarations

Competing Interests

The authors declare that they have no competing interests.

Author Contributions

Lei Zheng conceived and designed the experiments, analyzed the data, authored or reviewed drafts of the article, and approved the final draft.

Jia Lu performed the experiments, analyzed the data, prepared figures and/or tables, authored or reviewed drafts of the article, and approved the final draft.

Dalu Kong performed the experiments, analyzed the data, prepared figures and/or tables, and approved the final draft.

Yang Zhan conceived and designed the experiments, analyzed the data, authored or reviewed drafts of the article, and approved the final draft.

Data Availability

The following information was supplied regarding data availability:

The public dataset used in this study is available in GSE Collection H.

The raw data is available in GitHub and Zenodo:

- https://github.com/yangzhandr/Raw-data.git - yangzhandr. (2024). yangzhandr/Raw-data: Updated raw data (v.1.1.1). Zenodo. https://doi.org/10.5281/zenodo.14015463.

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
