# Peer review of "A gene signature related to programmed cell death to predict immunotherapy response and prognosis in colon adenocarcinoma"

_PeerJ, doi:10.7717/peerj.18895_

## Round 0.1 · original submission · Major Revisions

After careful review of your manuscript and consideration of the feedback from both reviewers, I believe your work shows promise but requires major revision. The reviewers have raised several substantive points that need to be addressed to strengthen the manuscript. Please provide a detailed response to all reviewers' comments along with your revised manuscript.

Reviewer 1 ·

Basic reporting

The aim of this study is to reveal cancer prognostic signature genes associated with programmed cell death features in colon cancer through bioinformatics analysis, and to construct a prognostic risk assessment model as a result. In this study, firstly, the information of colon cancer samples was obtained from public databases, and the molecular subtypes of colon cancer with different programmed cell death characteristics were elucidated by consensus clustering. The follow-up study focuses on mining signature genes associated with colon cancer prognosis and constructing a prognostic risk assessment model through COX regression analysis, and verifying the regulatory roles of these signature genes in the malignant phenotypes of cancer cells through joint cellular experiments. Finally, a column chart was constructed to verify the clinical value of the risk scores of the risk assessment model for colon cancer prognosis. In conclusion, this study is a comprehensive analysis, but the following issues still need to be addressed before publication:
1. This study's data acquisition for colon cancer was mainly realized through the TCGA dataset, but the subsequent construction of prognostic models will inevitably require a validation set, and through which database was this validation set acquired? Please complete this in the Methods section and reflect it in the Abstract section.
2. The methods section of this article needs further modification, for example, the culture of cells and the methods related to RT-qPCR should be differentiated so that the article is more clearly organized. It is recommended that this be revised.
3. Please note how the abbreviations in this paper are correctly cited, for example, the primer sequences contain both Reverse Primer and Forward Primer parts, but this study uses “R” and “F” directly without explaining them. This is clearly not standardized, and it is recommended that similar errors be checked and corrected in the full text.
4. What was the basis of this study to screen CDKN2A from the seven prognostic signature genes for subsequent cellular experiments? Based solely on the significant upregulation of the gene's expression in cancer cell samples is not convincing, and it is recommended that appropriate literature reports be added to support this in the results section.
5. Some of the statements in this study are too absolute, for example, it is not possible to determine that the poorer prognosis of patients with type S1 is caused by pathways related to tumor invasion and progression by enrichment analysis alone, and it is recommended that such statements be revised.
6. This study revealed high immune checkpoint gene expression and high TIDE scores in colon cancer patients in the high-risk score group, suggesting that the RiskScore is potentially associated with cancer immune escape and prediction of cancer immunotherapy response, and does not directly indicate poor immunotherapy response in the high-risk score group. It is recommended that the description of the results in this section be modified.
7. The Abstract section describes the results of the mutational landscape analysis performed in this study, elucidating the higher rate of TP53 somatic mutations in the high-risk scoring group, suggesting that this is one of the key findings of the paper. However, this study was not followed up with an in-depth discussion of the results, so please explain why and add a description if necessary.
8. It is suggested to discuss the prognostic characteristic genes of this study into two categories, one is prognostic protective factors while the other is prognostic risk factors, and at the same time, combine the relevant results of CIBERSORT and TIDE in this paper, to propose rational hypotheses on the immune mechanisms of these prognostic genes affecting cancer progression and programmed cell death, so as to make the content of this paper more informative.
9. An interesting finding of this study was the inhibition of fatty acid metabolic pathways in colon cancer samples from the high-risk scoring group, whereas it is well known that the activation of fatty acid metabolic pathways has a promotive effect on the malignant phenotype of cancer cells. Thus, it is recommended that this be rationalized in conjunction with other results and presented in the discussion section.
10. In this paper, Figure 6 illustrates that several immune checkpoint molecules showed a trend of upregulation in the high-risk scoring group, but conventional immune checkpoint molecules such as CD274 and CTLA4 did not show a significant difference between the high- and low-risk scoring groups, does it mean that the prognostic signature genes mined in this study could not accurately predict immune response in colon cancer? Please provide an explanatory note on this.

Experimental design

no comment

Validity of the findings

no comment

Reviewer 2 ·

Basic reporting

no comment

Experimental design

no comment

Validity of the findings

no comment

Additional comments

In this study, a programmed cell death-related RiskScore model was developed for prognosis of patients with colon adenocarcinoma. The experimental design is reasonable, the logic is clear, and the results support the conclusion. But I have some confusion that need to be confirmed by author:
1. What are the risk factors for colon cancer, and what are the symptoms of colon cancer. What are the main current options for treating colon cancer.
2. Line 48-50 (page 4), How is the 1254 PCD- related genes obtained, from gene set or previous study. Please supplement the source to achieve results reproduction.
3. In the results, before the description of each result, could the author please add the corresponding method and method part used in the result. Such as in line 31-32, the KM survival analysis revealed that the patients in S2 subtype showed a relatively favorable prognosis.
4. Line 52 (page 9), Is the space of “CDKN2 A” too large. Line 14-15 (page 10) of Figure 4H, What are the exact results.
5. Figure 7A, the author performed the somatic mutation analysis and found that the EMT, angiogenesis, and interferon-γ response were activated in high-risk group but fatty acid metabolism was inhibited. Usually, the EMT and angiogenesis are associated with the malignancy of the tumor, the interferon-γ response is associated with the anti-tumor response. This is an interesting result, please add appropriate discussion.
6. In addition, the mutation of TP53, TRIM46, MYO1G exhibited huge difference. How do these genes drive tumor development in various risk groups.
7. The Fatty acid metabolic pathway was inhibited in the high-risk group. How does it affect the malignancy of the tumor? Is there any report on that.
8. Line37-59 (page 12), the author listed others PCDscore system in colon cancer. What are the innovations or new findings in this paper compared with previous studies.
9. What prognostic models are used clinically. What is the clinical help of this article for the treatment of colon cancer
10. What is the prospect and significance of this article.

---

## Round 0.2 · accepted · Accept

I have reviewed your revised manuscript along with the reviewers' comments. Both reviewers have carefully evaluated your revisions and found that you have adequately addressed all their previous concerns. Based on their recommendations, I am happy to accept your manuscript for publication.

Reviewer 1 ·

Basic reporting

no comment

Experimental design

no comment

Validity of the findings

no comment

Additional comments

The purpose of this study is to reveal cancer prognostic marker genes associated with programmed cell death in cancer through bioinformatics analysis, and to construct a prognostic risk assessment model based on this. This study first obtained information on colon cancer samples from public databases, and through consistent clustering, elucidated cancer molecular subtypes with different programmed cell death characteristics. Subsequent research focused on exploring marker genes related to the prognosis of colon cancer, constructing a prognostic risk assessment model through COX regression analysis, and verifying the regulatory role of these marker genes in the malignant phenotype of cancer cells through joint cell experiments. The clinical value of the risk score of the final evaluation model for the prognosis of colon cancer. In summary, this study is a comprehensive analysis that has been thoroughly revised by the authors in response to the comments from the reviewers, and can be published in its current form.

Reviewer 2 ·

Basic reporting

Thank you for patiently responding to my comment. In this study, a programmed cell death related RiskScore model was developed for the prognosis of colon adenocarcinoma patients. The experimental design is reasonable, the logic is clear, and the results support the conclusion. There are no new comments now. Wishing editors and authors a happy New Year.

Experimental design

no comment

Validity of the findings

no comment